# Fabrication and Characterization of Polylactic Acid Electrospun Wound Dressing Modified with Polyethylene Glycol, Rosmarinic Acid and Graphite Oxide

**DOI:** 10.3390/nano13132000

**Published:** 2023-07-03

**Authors:** Chengyi Liu, Guicai Du, Qunqun Guo, Ronggui Li, Changming Li, Hongwei He

**Affiliations:** 1College of Life Sciences, Qingdao University, Qingdao 266071, China; m17806282245@163.com (C.L.); duguicai@qdu.edu.cn (G.D.); gqunqun@163.com (Q.G.); lrg@qdu.edu.cn (R.L.); 2Schneider Institute of Industrial Technology, School of Automation, Qingdao University, Qingdao 266071, China; 3Industrial Research Institute of Nonwovens & Technical Textiles, College of Textiles & Clothing, Qingdao University, Qingdao 266071, China

**Keywords:** PLA nanofiber, electrospinning, rosmarinic acid, wound healing

## Abstract

Polylactic acid (PLA) is a biodegradable polymer made from natural sources, and its electrospinning (e-spinning) nanofiber membrane doped with antibacterial ingredients is widely used in the field of medical dressings. In this research, 9 wt% of rosmarinic acid (RosA) and 0.04 wt% of graphite oxide (GO) with synergistic antibacterial activity were introduced into the e-spinning PLA precursor solution, and the obtained PLA nanofiber membrane showed good antibacterial properties and wound healing effects. At the same time, a nonionic amphiphilic polymer, polyethylene glycol (PEG), was also introduced into this system to improve the hydrophilicity of the e-spinning membrane for wound healing application. The morphological characterization showed the RosA/GO and PEG did not affect the e-spinning of PLA. The tests of mechanical performance and wettability demonstrated that PEG and RosA/GO incorporated in PLA have migrated easily to the surface of the fiber. The e-spun PLA/PEG/RosA/GO membrane showed good antibacterial activity and promoted initial wound healing quickly, which would be a promising application in wound dressing.

## 1. Introduction

The skin is the organ in the human body that has the greatest contact area with the outside world. It has functions that resist the external environmental stimuli, such as antimicrobial functions and regulation of the body temperature. The skin exposed to the environment is also the most vulnerable, incurring injuries such as burns, frostbite and cuts [1]. Wound healing is a continuous process, and the inflammation that is caused ordinarily by bacterial infections results in a decrease in the wound healing rate [2]. Wound dressings are important medical materials that promote wound healing. A qualified and efficient wound dressing needs to have the effect of reducing the risk of bacterial infection and controlling the occurrence of inflammation [3,4].

Electrospinning (e-spinning) has been a rapidly developing technique for preparing nanofiber non-woven membranes in the past three decades, which have a wide range of applications, such as filtration, environmental protection, energy, biomedicine, food packaging and other fields [5,6,7]. The electrospun (e-spun) membranes utilized in tissue engineering and wound dressings has become a hot topic in the research and development of e-spinning technology [8,9,10,11,12]. Compared with traditional wound dressings, the fiber mesh substrate composed of e-spun nanofibers has a smoother surface and structure close to the natural extracellular matrix (ECM), which is good for cell adhesion, encapsulation and the protection of various bioactive substances [13,14]. Some e-spun membranes of natural and synthetic polymers or their composites with biocompatibility and biodegradability have been employed as substrates for wound dressings [15,16]. PLA is a thermoplastic, high-modulus and high-strength polymer that can be produced from renewable resources such as corn and potatoes. It has attractive properties, including biocompatibility, biodegradability and harmless degradation products [17]. PLA e-spun fiber membranes could also be used in biomedical engineering [18,19], wound dressings [12,20,21], etc. Although PLA has antibacterial effects to some extent [22], as a wound dressing, better antibacterial properties are needed by adding antibacterial active substances such as nano-silver [23], ZnO [24], plant extracts [25,26,27,28] or antibiotics [29]. To improve the hydrophilicity of PLA e-spun membranes as wound dressings, both hydrophilic polymers and antibacterial active ingredients were introduced into the PLA e-spun membranes, such as chitosan and aloin [30], chitosan/starch/zinc oxide [31], hyperbranched polyglycerol and curcumin [32], beta cyclodextrin/canamaldehyde/[33], etc.

Due to its excellent optoelectronic properties, mechanical performance and large specific surface area, various graphene and its derivative products have emerged since it was discovered [34]. Graphene, oxide graphene (GO), reduced graphite oxide (rGO) and their derivatives have also attracted great attention in the biological community due to their excellent biocompatibility and low cytotoxicity [35]. Several kinds of antibacterial GO derivative or composite have been developed, such as, silver-modified GO [36], clarithromycin-loaded GO composite [37], ornidazole-loaded graphene [38] chlorophyllin functionalized GO [39]. Rosmarinic acid (RosA) is a natural polyphenolic acid compound with strong antioxidant properties, as well as antibacterial and anti-inflammatory properties [40]. RosA has demonstrated its important applications in pharmaceuticals, food, cosmetics, etc.

In this research, biodegradable polymer PLA was used as the e-spun nanofiber material matrix, and GO/RosA introduced as the synergistic antibacterial and anti-inflammatory active ingredient to study the effect of their addition on e-spinning of PLA and on the antibacterial synergistic effect of the e-spun composite membrane. At the same time, polyethylene glycol (PEG) was added as well, as a safe and biodegradable polymer to improve the hydrophilicity of PLA e-spun membranes. The performance characterization of the obtained PLA/PEG/RosA/GO nanofiber membrane and its evaluation on wound healing in mice as a wound dressing material, showed that the e-spun membrane could be employed as a biodegradable, absorbable and good antibacterial wound dressing.

## 2. Materials and Methods

### 2.1. Materials

Commercial polylactic acid (PLA) polymer, Ingeo^TM^ biopolymer 4032D, with a melt flow rate (MFR) = 7 g/10 min (210 °C, 2.16 kg) by NatureWorks (Minneapolis, MN, USA), was purchased from a distributor, Suzhou Ziyunxuan Plastic Co., Ltd. (Suzhou, China). Polyethylene glycol (PEG, molecular weight = 2000 g/mol), graphene oxide (2 mg/mL, dispersed in N-methyl pyrrolidone), rosmarinic acid (RosA), dichloromethane (DCM), N,N-dimethylformamide (DMF), sodium chloride, and sodium pentobarbital were all purchased from Sinopharm Chemical Reagent Co., Ltd. (Shanghai, China).

### 2.2. Preparation of e-Spinning Solutions

The 10% (*w*/*v*) solution of PLA and PLA/PEG (95:5 *w*/*w*) solution was prepared in DCM/DMF (8:2, *v*/*v*) on a magnetic stirrer for 2 h at room temperature, and RosA and GO was added in the solution with different content, as shown in Table 1. The e-spinning precursor solution and the e-spun nanofiber membranes were named after PLA, PLA/RosA/GO and PLA/PEG/RosA/GO, respectively.

### 2.3. Preparation of e-Spinning Membranes

The solutions shown in Table 1 were utilized for e-spinning on home-made e-spinning equipment with a syringe pump, high DC power source to produce nanofiber membranes (Figure 1). The parameters of the e-spinning process were set as follows: the voltage was 20 kV, 0.8 mL/h of feeding rate and 18 cm distance of tip-to-collector at room temperature (~20 °C) and about 50% humidity. The obtained fiber membranes were dried at 60 °C in an oven for 24 h. All samples were given with 1.2 g/m^2^ in basis weight.

### 2.4. Characterizations

#### 2.4.1. Morphology and Structure

The e-spun fibrous membranes were observed on a scanning electron microscope (SEM, Phenom Pro, Hitzacker, Germany) with a working distance of about 5 mm and an accelerator voltage of 15 kV under a high vacuum. The chemical structure of as-spun membranes was characterized by means of a Fourier transform infrared (FT-IR) spectroscopy (Nicolet iS10 instrument, Thermo Fisher Scientifc, Waltham, MA, USA) with ATR mode, scanning from 4000 to 400 cm^−1^, and X-ray diffraction (XRD) spectra were afforded by means of an X-ray diffractometer (DX-2700, Cephas, Taipei, Taiwan) with Cu Kα radiation, 40 kV of voltage, 40 mA of current, 5–50° of scanning range and 5°/min of scanning speed.

#### 2.4.2. Thermal and Mechanical Performance

The thermal properties were performed by using a TGA/DSC 3+ instrument (Mettler-Toledo, Germany) at a heating rate of 10 °C/min in a N_2_ atmosphere from 25 to 300 °C. In addition, the thermogravimetric analyzing (TGA) chart was also given from 25 °C to 600 °C at a rate of 10 °C/min in a N_2_ atmosphere.

The e-spun membrane’s mechanical properties were measured using a tensile tester (Instron 3382, Norwood, MA, USA). Before testing, different samples with a thickness of 0.20 ± 0.02 mm were cut 50 × 10 mm, and the obtained strips were clamped between tensile grips. The initial distance between the grips was 20 mm. The upper grip was then raised by a constant speed of 0.5 mm/s. The elongation at break was recorded together with the related tensile strength. Five strips of every sample were measured and the results were given with the average value.

#### 2.4.3. Hydrophilicity

The hydrophilicity of the as-spun membrane was characterized by the change of water contact angle, which was measured using a contact angle goniometer, JY-Phb (Chengde Jinhe Co., Ltd., Chengde, China). Three titrated water droplets dropping onto the e-spun membrane and the contact angle between the droplets and the membrane was observed and recorded.

#### 2.4.4. Porosity and Permeability

The pore size and its distribution of the e-spun membranes was tested on a pore-size tester (TOPAS PSM-165, Frankfurt, Germany) with the bubbling method, and isopropyl alcohol (IPA) was dripped and then completely infiltrated the sample (the tested area: 2.01 cm^2^). The porosity (P) was calculated via the following formula [41]:(1)$$\mathrm{Porosity}=1-\frac{m}{t\times A\times \rho }$$
where *m*, *t* and *A* are the mass, the thickness and the per unit membrane measured, respectively. And ρ is ~1.20 g/cm^3^, the density of raw materials, 1:1 of PLA/PEG.

The air permeability was detected by means of an Air Permeability Tester (FX3300, Textest AG, Switzerland) with a 20 cm^2^ effective area and a 200 Pa pressure drop.

### 2.5. Antibacterial Activity

According to the AATCC100-2004 standard, different antimicrobial tests were carried out with Escherichia coli (*E. coli*, ATCC 25922) and Staphylococcus aureus (*S. aureus*, ATCC 29213). A single colony of bacteria was inoculated in 5 mL LB liquid culture medium, and cultivated in a shaking incubator at 37 °C for 24 h to obtain a bacterial suspension with a concentration of 1 × 10^9^ CFUs/mL. Then it was diluted to 1 × 10^5^ CFUs/mL using sterile phosphate buffered saline (PBS). Forty mg of as-spun nanofiber membranes was added into 5 mL of diluted bacterial suspension, and another 5 mL of bacterial suspension as the control group. Cultivating in a shaker at 250 rpm and 37 °C for 12 h, 100 μL of bacterial suspension was coated onto an LB agar plate for bacterial counting.

### 2.6. In Vivo Wound Healing Study

Nine KM mice (20–25 g) were taken from Sibeifu Biotechnology Co., Ltd. (Suzhou, China). The dorsal region hairs of the anesthetized mice (pentobarbital sodium 60 mg/kg of body weight) were shaved, and a 6 mm circular excision diameter was cut from the back, then the wound was covered with an e-spun membrane patch. The mice were randomly distributed into 3 groups. The groups included a control based on pure e-spun PLA patch, PLA/RosA/GO and PLA/PEG/RosA/GO. On days 1, 3, 5,7, 9 and 11 of post-wounding, the photographs of wound sites were taken. Wound areas were measured through using the Image J software and compared with the original wound at each time interval for the rate of wound healing [1].

## 3. Results and Discussion

### 3.1. Characterization of e-Spun Nanofibers Morphology

As a biodegradable polyester, PLA’s e-spinning has been paid much attention recently and mixed solvents are used commonly for preparing e-spinning precursor solutions, such as CH_2_Cl_2_ and DMF [42,43,44]. In this work, 8:2 (*v*/*v*) of CH_2_Cl_2_ and DMF was employed; however, GO did not easily disperse in this system. Fortunately, the addition of RosA helped disperse the GO, which showed there might have been a redox that had occurred between the reducing RosA and GO. As shown in Appendix A, the color of the RosA/GO solution was darkening compared with the GO before the RosA added, which could illustrate that reduced GO (graphene) with a dark color was produced. To improve the wettability of the e-spinning membrane, the amphiphilic polymer PEG was introduced, which exhibited a more stable dispersive precursor solution throughout the entire spinning time of 2–4 h, or even over 10 h. As shown in Figure 2, the average diameter of the e-spinning fibers of pure PLA was 910 nm, and with the addition of RosA and GO, the average diameter of the fibers slightly decreased to 720 nm, mainly because the addition of GO (dispersion in NMP) reduced the concentration of the system. After adding PEG, the average diameter of the fibers decreased continuously to about 680 nm (Figure 2c), which may be attributed to it being a better dispersion of this e-spinning precursor solution.

### 3.2. Analysis of FT-IR and XRD for e-Spinning Fibers’ Structure

As shown in Figure 3a, the peaks of 1758 cm^−1^ and 1083 cm^−1^ were assigned to C=O and C-O ester bonds of PLA. Because of the two carboxyl and four hydroxyl groups of RosA, the peak at 1715 cm^−1^ and its broadening area could be attributed to COOH and the vinyl ester of RosA; the intensity and wavenumber were lower than those of PLA. The introduction of only 5% PEG had little effect on the infrared spectrum peak, and the 2884 cm^−1^ CH_2_ peak and 1080 cm^−1^ peaks overlapped with the corresponding peak of PLA. The lower content of GO was hard to find in these IR spectra. As shown in Figure 3b, the XRD plot shows the semi-crystalline polymer of PLA, where there was a broad peak at 22.68° [45]. The addition of RosA/GO resulted in a slight shift in the crystallization peak, and there was a diffraction that peak appeared at 16.63°, which may also indicate that RosA had a certain interaction with the PLA matrix and caused the change of diffraction peaks [46,47]. After adding PEG, the diffraction peaks were similar to PLA/RosA/GO, indicating that PEG/RosA/GO homogeneously mixed into the PLA matrix in an amorphous form, and had no significant effect on the crystallinity and crystal structure of composite nanofibers.

### 3.3. Thermal Properties

The thermal properties of the e-spun nanofiber membrane were analyzed on a machine of DSC-TG; these diagrams are shown in Figure 4. The TG of PLA was at 60–80 °C and the transit temperature at 100 °C might have been cold crystallization. Based on the DSC curves, the crystallinity of PLA, PLA/RosA/GO and PLA/PEG/RosA/GO could be calculated and was 45%, 45% and 46%, respectively, which coincided with the results shown in the XRD patterns. The melting point of pure PLA was 168.1 °C, and the addition of RosA/GO and PEG reduced its melting point to 160.4 °C and 159.2 °C, respectively. On the contrary, the TG curve showed that both RosA/GO and PEG had improved the heat resistance of PLA. The 5% of thermal weight loss temperature increased from 288 °C of pure PLA to 310 °C and 315 °C, respectively (Appendix A). The TG curve was smooth and there were no obvious small molecular weight loss stages, which also indicated that the addition of Rosa/GO and PEG containing multiple oxygen groups was thoroughly dispersed in the PLA and interacted with the PLA in some forms, such as hydrogen bonds to improve the heat resistance of the composite membrane materials.

### 3.4. Mechanical Performance

Figure 5 depicts the stress–strain curve of the e-spun membranes that evaluated their mechanical performance. The addition of RosA/GO significantly improved the breaking strength, which increased by about three times from 0.8 MPa for PLA to 2.6 MPa for PLA/Rosa/GO. However, the elongation at break decreased 60%. The improvements of strength might be attributed to the degree of the orientation that was increased because the diameter of PLA/RosA/GO and PLA/PEG/RosA/GO was finer than that of the pristine PLA. Accordingly, the number of fibers per unit weight or size of the former e-spun membranes was higher, which would be helpful to improve their strength [48,49]. The added nonionic amphiphilic PEG improved the dispersion and interaction of RosA/GO and PLA, which caused the crystallinity to increase, the fibers to become brittle and elongation at break to be reduced to 15%, while still maintaining high strength.

### 3.5. Wettability

Wettability is the main prerequisite to absorb wound exudate and keep the wound moist to some degree. As shown in Figure 6, PLA, as a hydrophobic material, had a water contact angle of approximately 128° for its e-spun membrane. When the hydrophilic RosA/GO was added, the contact angle decreased to 115°, but it was still not hydrophilic. The addition of the amphiphilic polymer PEG was very important and about 4.5 wt% was used, which exhibited a significant synergistic effect with RosA/GO; the water CA of the e-spun membrane was reduced to 74°. The similarly decreasing water CA (hydrophilicity) required the addition of about 20 wt% [50]; and in this work, PEG and RosA/GO might have easily migrated to the surface of the fiber. The e-spun membrane was hydrophilic and had similar wettability to conventional wound dressings [51].

### 3.6. Porosity and Permeability

Porosity and permeability are the other important performances for wound dressings, and they depend on the pore size and distribution. As shown in Figure 7, the PLA/RosA/GO and PLA/PEG/RosA/GO had 1.3 µm and 1.4 µm in average pore size, respectively, and lower porosity, lower than that of PLA due to larger diameter of the PLA e-spun fibers. However, the permeability of these membranes was about 300 mm/s, which was not much difference from each other. PLA/PEG/RosA/GO would be more suitable for wound dressing due to having better wettability.

### 3.7. Evaluation of Antimicrobial Properties

As shown in Figure 8, the Gram-positive bacteria, *Staphylococcus aureus* and the Gram-negative bacteria, *Escherichia coli* were used as indicators to conduct antibacterial tests on as-spun nanofiber membranes. The pristine PLA showed no antibacterial activity according to this evaluation method because there was no antibacterial composition migrated. After the antibacterial composition of the RosA/GO was introduced, some of the RosA/GO distributed on the surface of fiber came off, which resulted in the number of colonies decreasing significantly. Especially when the PEG was introduced, the sample was almost sterile, which indicated further that the hydrophilic PEG helped more antibacterial RosA/GO migrate out of the fiber and dissolve in the PBS liquid. At the same time, RosA/GO showed better bactericidal activity against *S. aureus*, which also coincided with the early evaluated results that rosmarinic acid had good antibacterial and anti-inflammatory activity [52]. These showed that as-obtained PLA/PEG/RosA/GO was more profitable for the wound dressing application because *S. aureus* was the most common bacteria causing wound inflammation [40].

### 3.8. In Vivo Wound Healing Studies

The outcomes of the in vivo wound healing experiments can be seen on days 1, 3, 5, 7, 9 and 11 for the different samples in Figure 9. Compared with the control (pure PLA), both the PLA/RosA/GO and PLA/PEG/RosA/GO patches showed better wound healing, and the latter did not differ significantly in the wound healing rate (Figure 10 and Appendix A). In fact, the patch of PLA/PEG/RosA/GO initially showed better adhesion on the surface of skin and absorbed liquid quickly due to its good wettability, which would help the wound healing and prevent the initial infection.

## 4. Conclusions

In the present study, a natural biodegradable polymer, PLA, was employed as a polymer matrix to electrospin nonfiber membranes. RosA and GO were introduced as antibacterial ingredients, and PEG was used as a hydrophilic modifier. The effect of RosA/GO and PLG was evaluated on the morphology and performances of the e-spun membranes. The strength enhancement and the lowering of the elongation at break was attributed to the RosA/GO promoting crystallization during fiber forming. The good wettability and the antibacterial activity of PLA/PEG/RosA/GO also demonstrated that there was a synergistic effect among PEG, RosA/GO and PLA; for example, PEG helping RosA/GO migrant easily to the surface of the fiber. Animal experiments showed that the e-spun membrane had good promotion on the initial wound healing, which would be a potential application in wound dressing.

## Figures and Tables

**Figure 1 nanomaterials-13-02000-f001:**
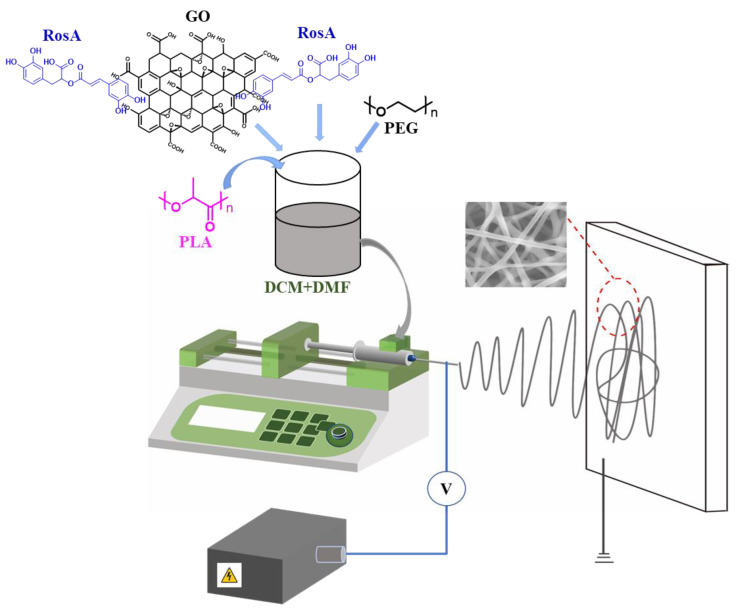
Illustration of a home-made e-spinning equipment.

**Figure 2 nanomaterials-13-02000-f002:**
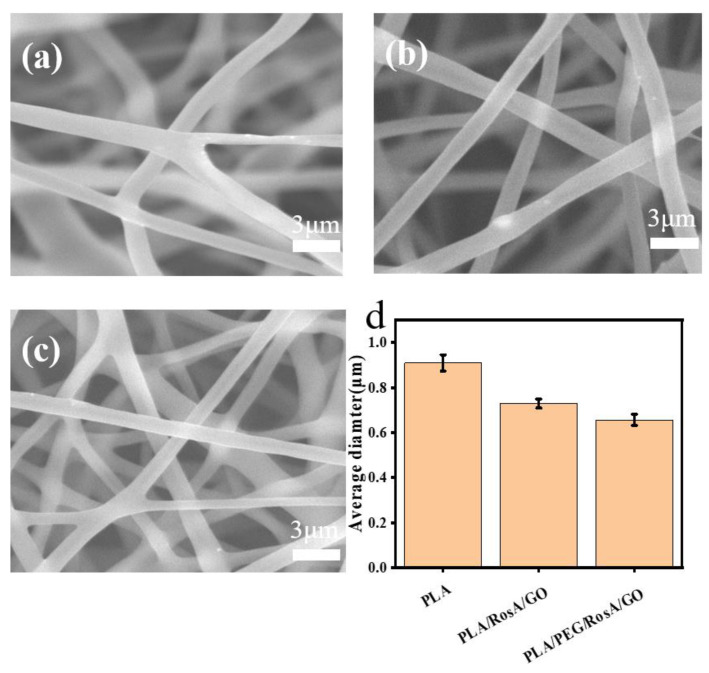
SEM images of electrospun composite nanofiber membrane: (**a**) pure PLA, (**b**) PLA/RosA/GO, (**c**) PLA/PEG/RosA/GO, (**d**) average diameter of fibers (**a**–**c**).

**Figure 3 nanomaterials-13-02000-f003:**
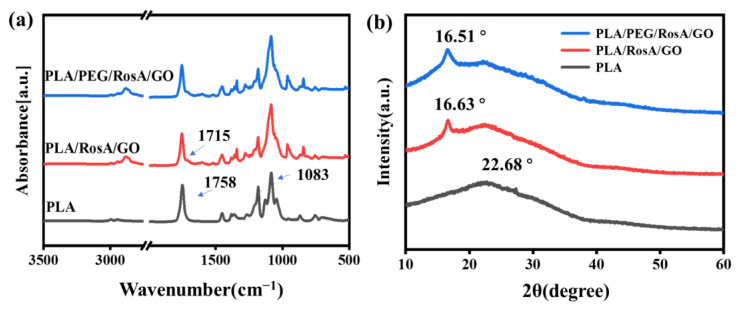
(**a**) FT-IR spectra and (**b**) XRD patterns of as-spun composite nanofiber membranes.

**Figure 4 nanomaterials-13-02000-f004:**
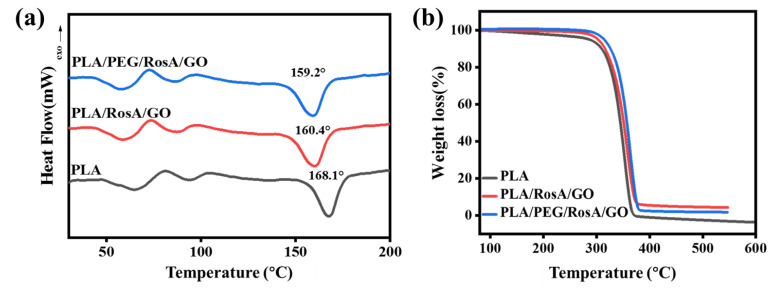
(**a**) DSC and (**b**) TG curves of as-spun composite nanofiber.

**Figure 5 nanomaterials-13-02000-f005:**
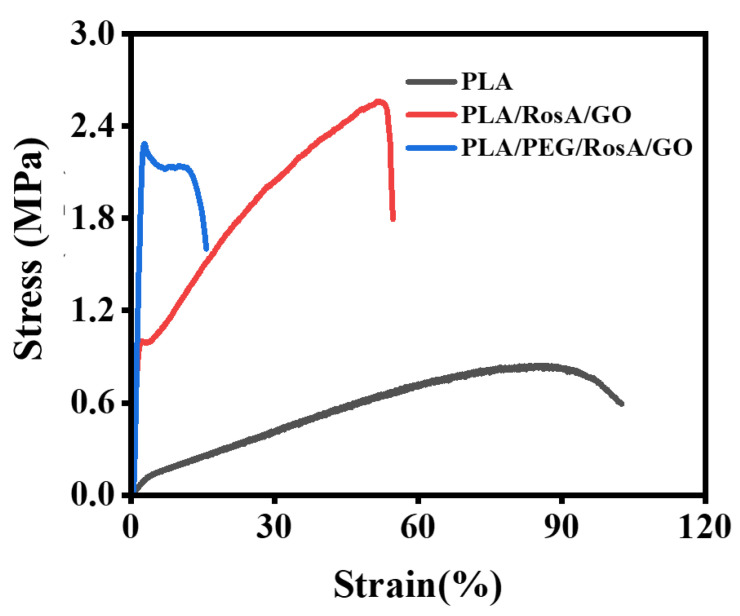
Stress-strain curve of the e-spun nanofiber membranes.

**Figure 6 nanomaterials-13-02000-f006:**
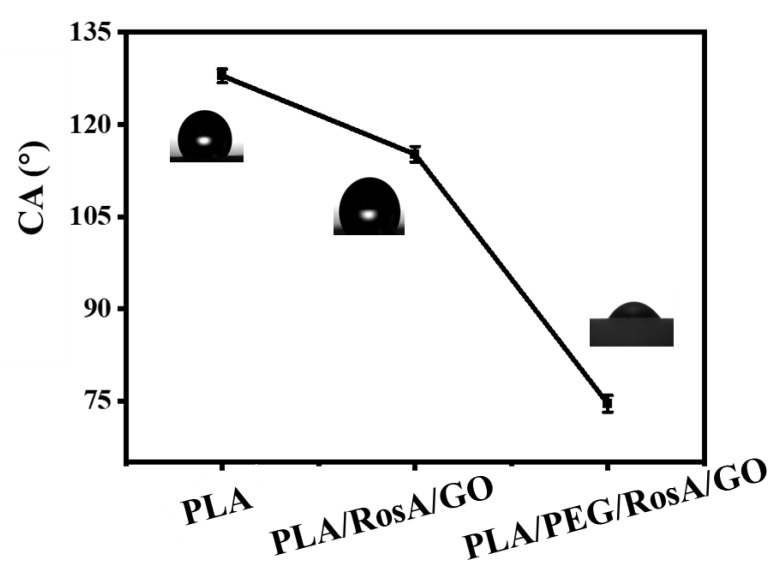
Water contact angle (CA) of the e-spun nanofiber membranes.

**Figure 7 nanomaterials-13-02000-f007:**
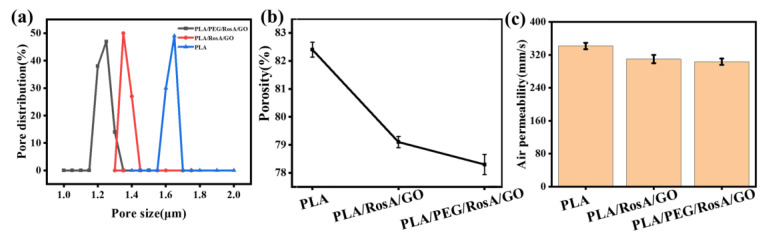
(**a**) Pore distribution, (**b**) porosity and (**c**) permeability of the e-spun nanofiber membranes.

**Figure 8 nanomaterials-13-02000-f008:**
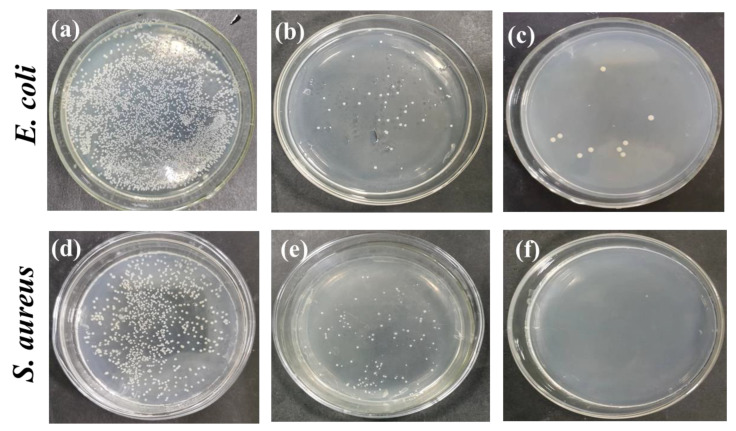
Evaluation of antibacterial activity of as-spun membranes: (**a**,**d**) pure PLA; (**b**,**e**) PLA/RosA/GO; (**c**,**f**) PLA/PEG/RosA/GO.

**Figure 9 nanomaterials-13-02000-f009:**
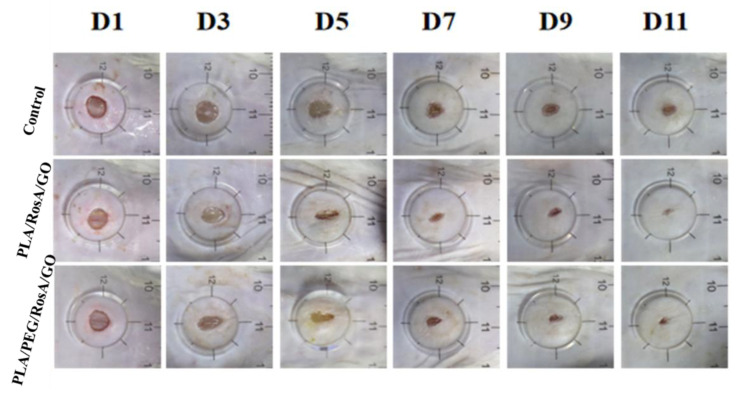
Photos of the status of wound healing for different membranes at 1, 3, 5, 7, 9, 11 days.

**Figure 10 nanomaterials-13-02000-f010:**
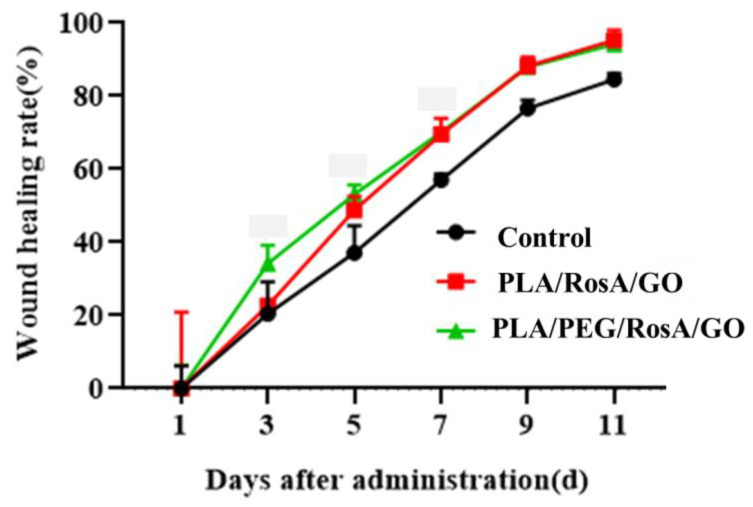
Healing rate of different treatment groups.

**Table 1 nanomaterials-13-02000-t001:** Recipe of e-spinning solution.

Run.	DCM/DMF, mL/mL	PLA, g	PEG, g	RosA, g	GO, mL
PLA	8/2	1	0	0	0
PLA/RosA/GO	8/2	1	0	0.1	2
PLA/PEG/RosA/GO	8/2	0.95	0.05	0.1	2

## Data Availability

The data that support the findings of this research are available from the corresponding authors on reasonable request.

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
