# Peer review of "Fabrication and Characterization of Polylactic Acid Electrospun Wound Dressing Modified with Polyethylene Glycol, Rosmarinic Acid and Graphite Oxide"

_nanomaterials, 2023, doi:10.3390/nano13132000_

Round 1
Reviewer 1 Report
It should be noted right away that the results presented in the article on the development of an antibacterial coating based on PLA, polyethylene glycol (PEG), rosmarinic acid (RosA) and graphite oxide (GO) have little prospect of being implemented in practice as wound dressings. Indeed, Figure 10 demonstrates that the introduction of all the above-mentioned fillers into the PLA matrix does not lead to any significant effect on wound healing.
Nevertheless, the article may be of interest to specialists in the field of electrospinning polymers, as well as composites based on them.
Some comments on the text of the article:
1) Line 61. It is advisable to provide a link to the article where the antibacterial activity of PLA is noted.
2) Line 70. « oxide graphene oxide (GO)» is probably a typo.
3) Line 195. Where is Figure.S1 in the article and how is redox proved between RosA and GO?
4) Line 203-204. Incomprehensible phrase: «After adding PEG, the average diameter of the fibers did not change much, and the nanofibers was much thinner». So, the diameter of the fiber has changed or not?
5) Figure 3b (XRD patterns). According to the XRD patterns presented in this figure, PLA is in an amorphous state, and not in a semi crystalline state. The appearance of the 16.63° reflex on the pattern is not necessarily since «RosA has a certain interaction with the PLA matrix» (Line 218), but it may be due to the crystal structure of graphene.
6) Figure 4a. What are the transitions in the temperature range of 50-100°C? It is desirable to clarify the degree of crystallinity of PLA according to the DSC data and whether it coincides with the X-ray diffraction data (Figure 3b)?
7) Line 242-246. The authors argue that «The addition of RosA/GO significantly improved the breaking strength…» due to the fact that «RosA/GO played a role as a crystal nucleus during fiber forming and improved the crystallinity of the fiber». However, the data of Figures 4a and, moreover, Figure 3b do not confirm that the crystallinity of PLA fibers filled with RosA/GO is higher than that of unfilled PLA fibers.
Author Response
Dear reviewer,
Thank you very much for your hard work of reviewing this article.
The response was given point by point in a Word file attached, and all revisions highlighted in the revised manuscript. Please check it.
Thanks a lot.
Regards
The authors.

Reviewer 2 Report
Major Revision required
1) Is it rosemary acid or rosmarinic acid? Line no 16, it is written rosemary, and in other places it is mentioned as rosmarinic acid.
2) To verify the applications of the electrospinning technique in the biomedical field, including wound dressing, some latest research should be cited. Some example-
https://doi.org/10.3390/pharmaceutics11070305
https://doi.org/10.3390/pharmaceutics15051347
3) Please, be consistent while writing the electrospinning parameters. (the applying voltage and distance from the tip to the collector in section 2.3 ).
4) What were the viscosity and conductivity of the prepared solutions?
5) There is no evidence to prove the presence of GO in the fibers. (SEM, IR, and XRD do not show GO in the fiber).
6) The antibacterial results should be explained in detail. What is the main factor for antibacterial property here? In my opinion, the phenolic groups present in the RosA also play a role in antibacterial performance. The authors can get further information from the below reference.
- https://doi.org/10.1080/00914037.2017.1376200
7) Why is the antibacterial action pronounced higher in S. aureus than E. coli?
8) The authors should check this manuscript carefully and fix the grammar/punctuation errors.
Author Response

(The authors gave the same response as above.)

Reviewer 3 Report
The paper reports new PLA based fibers incorporating rosmarinic acid, GO, and/or not PEO. It is a potential interesting manuscript, but at this stage there are major concerns that need to be addressed.
1. The goal of the manuscript needs to be clearly stated.
2. The receipt for preparation of the electrospinning solutions will be carefully revised for accuracy, as there are big discrepancies between the data given in table 1 and those given in the text, e.g. in the abstract it was stated that rosmarinic acid acid was in concentration of 9 wt%, but in table 1 it appeared it was around 50%; in the text it was stated that PLA solution was 10%, but from the table 1 it appeared it was 1%, an so on.
3. From the experimental part it is not clear what kind of fibers were prepared, core/shell fibers via biaxial electrospinning? What is the shell and what is the core? Why two electrospinning methods were described? It is total confusing, the authors should revise this part.
4. The data will be statistically treated in order to clearly assess if the differences in properties are induced by the different composition.
5. Comparison with other PLA nanofibers will be done in order to highlight the advantages and limitations of the studies fibers.
6. Apparently, there are no differences in the wound healing effect between the sample containing PEO vs. the sample without PEO. This should be commented. Further, the role of the rosmarinic acid and GO on wound healing will be commented too.
7. A cytotoxicity test should be done, in order to assess the lack of toxicity of the studied samples.
8. The English language will be carefully revised, because often is affecting the scientific meaning, e.g.: „easy to migrant”; „could be attributed to the redox between reducing RosA and GO to a certain extent”; „the color of the RosA/GO solution was darker than GO, which could illustrate that deeper color reduced GO (Graphene) was produced”; „the average diameter of the fibers did not change much, and the nanofibers was much thinner”- if the nanofibers were much thinner, this means that actualy the average diameter change significantly!?; „molecure of RosA”; „the strength and wavenumber” should be „the intensity and wavenumber”.
The English language will be carefully revised, because often is affecting the scientific meaning, e.g.: „easy to migrant”; „could be attributed to the redox between reducing RosA and GO to a certain extent”; „the color of the RosA/GO solution was darker than GO, which could illustrate that deeper color reduced GO (Graphene) was produced”; „the average diameter of the fibers did not change much, and the nanofibers was much thinner”- if the nanofibers were much thinner, this means that actualy the average diameter change significantly!?; „molecure of RosA”; „the strength and wavenumber” should be „the intensity and wavenumber”.
Author Response

(The authors gave the same response as above.)

Round 2
Reviewer 2 Report
The paper can be accepted in its current form.
Reviewer 3 Report
The article has been somehow improved, still needs extensive English editing, including many scientific statements.
English needs extensive English revision, there are many inaccurate scientific statements, e.g. "showed there might be a redox occurred" it should be "showed there might be a redox reaction occurred"; "transit temperature" should be "transition temperature", and so on.